# LEARNING SPACE PARTITIONS FOR NEAREST NEIGHBOR SEARCH

**Yihe Dong**[*]
Microsoft

**Piotr Indyk**
MIT

**Ilya Razenshteyn**
Microsoft Research

**Tal Wagner**
MIT

## ABSTRACT

Space partitions of $\mathbb{R}^d$ underlie a vast and important class of fast nearest neighbor search (NNS) algorithms. Inspired by recent theoretical work on NNS for general metric spaces (Andoni et al., 2018b;c), we develop a new framework for building space partitions reducing the problem to *balanced graph partitioning* followed by *supervised classification*. We instantiate this general approach with the KaHIP graph partitioner (Sanders & Schulz, 2013) and neural networks, respectively, to obtain a new partitioning procedure called *Neural Locality-Sensitive Hashing (Neural LSH)*. On several standard benchmarks for NNS (Aumüller et al., 2017), our experiments show that the partitions obtained by Neural LSH consistently outperform partitions found by quantization-based and tree-based methods as well as classic, data-oblivious LSH.

## 1 Introduction

The *Nearest Neighbor Search (NNS)* problem is defined as follows. Given an $n$-point dataset $P$ in a $d$-dimensional Euclidean space $\mathbb{R}^d$, we would like to preprocess $P$ to answer $k$-nearest neighbor queries quickly. That is, given a query point $q \in \mathbb{R}^d$, we want to find the $k$ data points from $P$ that are closest to $q$. NNS is a cornerstone of the modern data analysis and, at the same time, a fundamental geometric data structure problem that led to many exciting theoretical developments over the past decades. See, e.g., Wang et al. (2016); Andoni et al. (2018a) for an overview.

The main two approaches to constructing efficient NNS data structures are *indexing* and *sketching*. The goal of indexing is to construct a data structure that, given a query point, produces a small subset of $P$ (called *candidate set*) that includes the desired neighbors. Such a data structure can be stored on a single machine, or (if the data set is very large) distributed among multiple machines. In contrast, the goal of sketching is to compute compressed representations of points to enable computing approximate distances quickly (e.g., compact binary hash codes with the Hamming distance used as an estimator, see the surveys Wang et al. (2014; 2016)). Indexing and sketching can be (and often are) combined to maximize the overall performance (Wu et al., 2017; Johnson et al., 2017).

Both indexing and sketching have been the topic of a vast amount of theoretical and empirical literature. In this work, we consider the *indexing* problem. In particular, we focus on indexing based on *space partitions*. The overarching idea is to build a partition of the ambient space $\mathbb{R}^d$ and split the dataset $P$ accordingly. Given a query point $q$, we identify the bin containing $q$ and form the resulting list of candidates from the data points residing in the same bin (or, to boost the accuracy, nearby bins as well). Some of the popular space partitioning methods include locality-sensitive hashing (LSH) (Lv et al., 2007; Andoni et al., 2015; Dasgupta et al., 2017); quantization-based approaches, where partitions are obtained via $k$-means clustering of the dataset (Jégou et al., 2011; Babenko & Lempitsky, 2012); and tree-based methods such as random-projection trees or PCA trees (Sproull, 1991; Bawa et al., 2005; Dasgupta & Sinha, 2013; Keivani & Sinha, 2018).

Compared to other indexing methods, space partitions have multiple benefits. First, they are naturally applicable in *distributed* settings, as different bins can be stored on different machines (Bahmani et al., 2012; Ni et al., 2017; Li et al., 2017; Bhaskara & Wijewardena, 2018). Moverover, the computational efficiency of search can be further improved by using any nearest neighbor search algorithm locally on each machine. Second, partition-based indexing is particularly suitable for GPUs due to the simple and predictable memory access pattern (Johnson et al., 2017). Finally, partitions can be combined with cryptographic techniques to yield efficient *secure* similarity search algorithms (Chen et al., 2019). Thus, in this paper we focus on designing space partitions that optimize the trade-off between their key metrics: the number of reported candidates, the fraction of the true nearest neighbors among the candidates, the number of bins, and the computational efficiency of the point location.

---

[*]Author names are ordered alphabetically.

Recently, there has been a large body of work that studies how modern machine learning techniques (such as neural networks) can help tackle various classic algorithmic problems (a partial list includes Mousavi et al. (2015); Baldassarre et al. (2016); Bora et al. (2017); Dai et al. (2017); Metzler et al. (2017); Kraska et al. (2018); Balcan et al. (2018); Lykouris & Vassilvitskii (2018); Mitzenmacher (2018); Purohit et al. (2018)). Similar methods—under the name "learn to hash"—have been used to improve the *sketching* approach to NNS (Wang et al., 2016). However, when it comes to *indexing*, while some unsupervised techniques such as PCA or $k$-means have been successfully applied, the full power of modern tools like neural networks has not yet been harnessed. This state of affairs naturally leads to the following general question: **Can we employ modern (supervised) machine learning techniques to find good space partitions for nearest neighbor search?**

## 1.1 Our contribution

In this paper we address the aforementioned challenge and present a new framework for finding high-quality space partitions of $\mathbb{R}^d$. Our approach consists of three major steps:

1. Build the $k$-NN graph $G$ of the dataset by connecting each data point to $k$ nearest neighbors;

2. Find a balanced partition $\mathcal{P}$ of the graph $G$ into $m$ parts of nearly-equal size such that the number of edges between different parts is as small as possible;

3. Obtain a partition of $\mathbb{R}^d$ by training a classifier on the data points with labels being the parts of the partition $\mathcal{P}$ found in the second step.

See Figure 1 for illustration. The new algorithm *directly optimizes* the performance of the partition-based nearest neighbor data structure. Indeed, if a query is chosen as a uniformly random *data point*, then the average $k$-NN accuracy is exactly equal to the fraction of edges of the $k$-NN graph $G$ whose endpoints are separated by the partition $\mathcal{P}$. This generalizes to out-of-sample queries provided that the query and dataset distributions are close, and the test accuracy of the trained classifier is high.

At the same time, our approach is directly related to and inspired by recent theoretical work (Andoni et al., 2018b;c) on NNS for general metric spaces. In particular, using the framework of (Andoni et al., 2018b;c), we prove that, under mild conditions on the dataset $P$, the $k$-NN graph of $P$ can be partitioned with a hyperplane into two parts of comparable size such that only few edges get split by the hyperplane. This gives a partial theoretical justification of our method.

The new framework is very flexible and uses partitioning and learning in a black-box way. This allows us to plug various models (linear models, neural networks, etc.) and explore the trade-off between the quality and the algorithmic efficiency of the resulting partitions. We emphasize the importance of *balanced* partitions for the indexing problem, where all bins contain roughly the same number of data points. This property is crucial in the distributed setting, since we naturally would like to assign a similar number of points to each machine. Furthermore, balanced partitions allow tighter control of the number of candidates simply by varying the number of retrieved parts. Note that a priori, it is unclear how to partition $\mathbb{R}^d$ so as to induce balanced bins of a given dataset. Here the combinatorial portion of our approach is particularly useful, as balanced graph partitioning is a well-studied problem, and our supervised extension to $\mathbb{R}^d$ naturally preserves the balance by virtue of attaining high training accuracy.

We speculate that the new method might be potentially useful for solving the NNS problem for *non-Euclidean* metrics, such as the edit distance (Zhang & Zhang, 2017) or optimal transport distance (Kusner et al., 2015). Indeed, for any metric space, one can compute the $k$-NN graph and then partition it. The only step that needs to be adjusted to the specific metric at hand is the learning step.

Let us finally put forward the challenge of scaling our method up to billion-sized or even larger datasets. For such scale, one needs to build an *approximate* $k$-NN graph as well as using graph partitioning algorithms that are faster than KaHIP. We leave this exciting direction to future work. For the current experiments (datasets of size $10^6$ points), preprocessing takes several hours. Another important challenge is to obtain NNS algorithms based on the above partitioning with *provable* guarantees in terms of approximation and running time. However, we expect it to be difficult, in particular, since all the current state-of-the-art NNS algorithms lack such guarantees (e.g., $k$-means-based (Jégou et al., 2011) or graph methods (Malkov & Yashunin, 2018), see also (Aumüller et al., 2017) for a recent SOTA survey).

**Evaluation** We instantiate our framework with the KaHIP algorithm (Sanders & Schulz, 2013) for the graph partitioning step, and either linear models or small-size neural networks for the learning step. We evaluate it on several standard benchmarks for NNS (Aumüller et al., 2017) and conclude that in terms of quality of the resulting partitions, it consistently outperforms quantization-based and tree-based partitioning procedures, while maintaining comparable

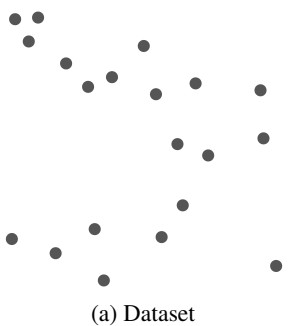 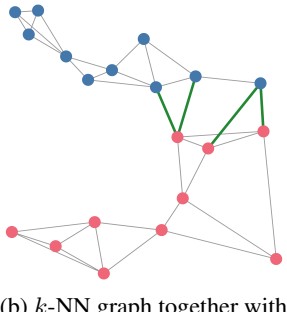 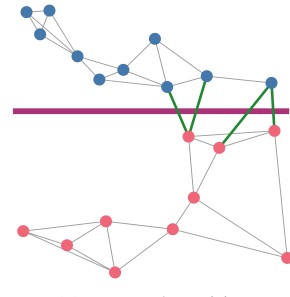

(a) Dataset          (b) $k$-NN graph together with          (c) Learned partition
                         a balanced partition

Figure 1: Stages of our framework

algorithmic efficiency. In the high accuracy regime, our framework yields partitions that lead to processing up to $2.3\times$ fewer candidates than the strongest baseline.

As a baseline method we use $k$-means clustering (Jégou et al., 2011). It produces a partition of the dataset into $k$ bins, in a way that naturally extends to all of $\mathbb{R}^d$, by assigning a query point $q$ to its nearest centroid. (More generally, for multi-probe querying, we can rank the bins by the distance of their centroids to $q$). This simple scheme yields very high-quality results for indexing. Besides $k$-means, we evaluate LSH (Andoni et al., 2015), ITQ (Gong et al., 2013), PCA tree (Sproull, 1991), RP tree (Dasgupta & Sinha, 2013), and Neural Catalyzer (Sablayrolles et al., 2019).

## 1.2 Related work

On the empirical side, currently the fastest indexing techniques for the NNS problem are *graph-based* (Malkov & Yashunin, 2018). The high-level idea is to construct a graph on the dataset (it can be the $k$-NN graph, but other constructions are also possible), and then for each query perform a walk, which eventually converges to the nearest neighbor. Although very fast, graph-based approaches have suboptimal "locality of reference", which makes them less suitable for several modern architectures. For instance, this is the case when the algorithm is run on a GPU (Johnson et al., 2017), or when the data is stored in external memory (Sun et al., 2014) or in a distributed manner (Bahmani et al., 2012; Ni et al., 2017). Moreover, graph-based indexing requires many rounds of adaptive access to the dataset, whereas partition-based indexing accesses the dataset in one shot. This is crucial, for example, for nearest neighbor search over encrypted data (Chen et al., 2019). These benefits justify further study of partition-based methods.

Machine learning techniques are particularly useful for the *sketching* approach, leading to a vast body of research under the label "learning to hash" (Wang et al., 2014; 2016). In particular, several recent works employed neural networks to obtain high-quality sketches (Liong et al., 2015; Sablayrolles et al., 2019). The fundamental difference from our work is that sketching is designed to speed up *linear scans* over the dataset, by reducing the *cost* of distance evaluation, while indexing is designed for *sublinear time* searches, by reducing the *number* of distance evaluations. We note that while sketches are not designed for indexing, they can be used for that purpose, since a $b$-bit hashing scheme induces a partition of $\mathbb{R}^d$ into $2^b$ parts. Nonetheless, our experiments show that partitions induced by these methods (such as Iterative Quantization (Gong et al., 2013)) are not well-suited for indexing, and underperform compared to quantization-based indexing, as well as to our methods.

We highlight in particular the recent work of Sablayrolles et al. (2019), which uses neural networks to learn a mapping $f\colon \mathbb{R}^d \to \mathbb{R}^{d'}$ that improves the geometry of the dataset and the queries to facilitate subsequent sketching. It is natural to ask whether the same family of maps can be applied to enhance the quality of *partitions* for indexing. However, as our experiments show, in the high accuracy regime the maps learned using the algorithm of Sablayrolles et al. (2019) consistently degrade the quality of partitions.

Finally, we mention that here is some prior work on learning space partitions: Cayton & Dasgupta (2007); Ram & Gray (2013); Li et al. (2011). However, all these algorithms learn *hyperplane* partitions into two parts (then applying them recursively). Our method, on the other hand, is much more flexible, since neural networks allow us to learn a much richer class of partitions.

## 2   Our method

Given a dataset $P \subseteq \mathbb{R}^d$ of $n$ points, and a number of bins $m > 0$, our goal is to find a partition $\mathcal{R}$ of $\mathbb{R}^d$ into $m$ bins with the following properties:

1. *Balanced:* The number of data points in each bin is not much larger than $n/m$.
2. *Locality sensitive:* For a typical query point $q \in \mathbb{R}^d$, most of its nearest neighbors belong to the same bin of $\mathcal{R}$. We assume that queries and data points come from similar distributions.
3. *Simple:* The partition should admit a compact description and, moreover, the point location process should be computationally efficient. For example, we might look for a space partition induced by hyperplanes.

Formally, we want the partition $\mathcal{R}$ that minimizes the loss $\mathbb{E}_q \left[ \sum_{p \in N_k(q)} \mathbf{1}_{\mathcal{R}(p) \neq \mathcal{R}(q)} \right]$ s.t. $\forall_{p \in P} \; |\mathcal{R}(p)| \leq (1 + \eta)(n/m)$, where $q$ is sampled from the query distribution, $N_k(q) \subset P$ is the set of its $k$ nearest neighbors in $P$, $\eta > 0$ is a balance parameter, and $\mathcal{R}(p)$ denotes the part of $\mathcal{R}$ that contains $p$.

First, suppose that the query is chosen as a *uniformly random data point*, $q \sim P$. Let $G$ be the $k$-NN graph of $P$, whose vertices are the data points, and each vertex is connected to its $k$ nearest neighbors. Then the above problem boils down to partitioning vertices of the graph $G$ into $m$ bins such that each bin contains roughly $n/m$ vertices, and the number of edges crossing between different bins is as small as possible (see Figure 1(b)). This *balanced graph partitioning* problem is extremely well-studied, and there are available combinatorial partitioning solvers that produce very high-quality solutions. In our implementation, we use the open-source solver KaHIP (Sanders & Schulz, 2013), which is based on a sophisticated local search.

More generally, we need to handle out-of-sample queries, i.e., which are not contained in $P$. Let $\widetilde{\mathcal{R}}$ denote the partition of $G$ (equivalently, of the dataset $P$) found by the graph partitioner. To convert $\widetilde{\mathcal{R}}$ into a solution to our problem, we need to extend it to a partition $\mathcal{R}$ of the whole space $\mathbb{R}^d$ that would work well for query points. In order to accomplish this, we train a model that, given a query point $q \in \mathbb{R}^d$, predicts which of the $m$ bins of $\widetilde{\mathcal{R}}$ the point $q$ belongs to (see Figure 1(c)). We use the dataset $P$ as a training set, and the partition $\widetilde{\mathcal{R}}$ as the labels – i.e., each data point is labeled with the ID of the bin of $\widetilde{\mathcal{R}}$ containing it. The method is summarized in Algorithm 1. The geometric intuition for this learning step is that – even though the partition $\widetilde{\mathcal{R}}$ is obtained by combinatorial means, and in principle might consist of ill-behaved subsets of $\mathbb{R}^d$ – in most practical scenarios, we actually expect it to be close to being induced by a simple partition of the ambient space. For example, if the dataset is fairly well-distributed on the unit sphere, and the number of bins is $m = 2$, a balanced cut of $G$ should be close to a hyperplane.

The choice of model to train depends on the desired properties of the output partition $\mathcal{R}$. For instance, if we are interested in a hyperplane partition, we can train a linear model using SVM or regression. In this paper, we instantiate the learning step with both *linear models* and *small-sized neural networks*. Here, there is natural tension between the size of the model we train and the accuracy of the resulting classifier, and hence the quality of the partition we produce. A larger model yields better NNS accuracy, at the expense of computational efficiency. We discuss this in Section 3.

**Multi-probe querying** Given a query point $q$, the trained model can be used to assign it to a bin of a partition $\mathcal{R}$, and search for nearest neighbors within the data points in that part. In order to achieve high search accuracy, we actually train the model to predict *several* bins for a given query point, which are likely to contain nearest neighbors. For neural networks, this can be done naturally by taking several largest outputs of the last layer. By searching through more bins (in the order of preference predicted by the model) we can achieve better accuracy, allowing for a trade-off between computational resources and accuracy.

**Hierarchical partitions** When the required number of bins $m$ is large, in order to improve the efficiency of the resulting partition, it pays off to produce it in a hierarchical manner. Namely, we first find a partition of $\mathbb{R}^d$ into $m_1$ bins, then recursively partition each of the bins into $m_2$ bins, and so on, repeating the partitioning for $L$ levels. The total number of bins in the overall partition is $m = m_1 \cdot m_2 \cdot \ldots m_L$. See Figure 2 for illustration. The advantage of such a hierarchical partition is that it is much simpler to navigate than a one-shot partition with $m$ bins.

**Neural LSH with soft labels** In the primary instantiation of our framework, we set the supervised learning component to a a neural network with a small number of layers and constrained hidden dimensions (the exact parameters are specified in the next section). In order to support effective multi-probe querying, we need to infer not just the bin that contains the query point, but rather a *distribution* over bins that are likely to contain this point and its neighbors. A $T$-probe candidate list is then formed from all data points in the $T$ most likely bins. In order to accomplish this, we use *soft labels* for data points generated as follows. For $S \geq 1$ and a data point $p$, the soft label $\mathcal{P} = (p_1, p_2, \ldots, p_m)$ is

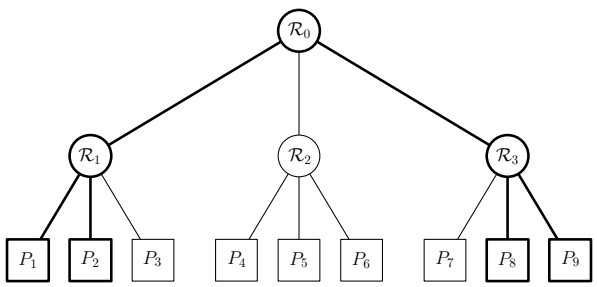

Figure 2: Hierarchical partition into 9 bins with $m_1 = m_2 = 3$. $\mathcal{R}_i$'s are partitions, $P_j$'s are the bins of the dataset. Multi-probe query procedure, which descends into 2 bins, may visit the bins marked in bold.

---

**Preprocessing**
Input: Dataset $P \subset \mathbb{R}^d$, integer parameter $k > 0$, number of bins $m > 0$

1: Build a $k$-NN graph $G$ of $P$.
2: Run a balanced graph partitioning algorithm on $G$ into $m$ parts. Number the parts arbitrarily as $1, \ldots, m$. Let $\pi(p) \in \{1, \ldots, m\}$ denote the part containing $p$, for every $p \in P$.
3: Train a machine learning model $M$ with training set $P$ and labels $\{\pi(p)\}_{p \in P}$. For every $x \in \mathbb{R}^d$, let $M(x) \in \{1, \ldots, m\}$ denote the prediction of $M$ on $x$.

$M(\cdot)$ defines our $m$-way partition of $\mathbb{R}^d$. Note that it is possible that $\pi(p) \neq M(p)$ for some $p \in P$, if $M$ attains imperfect training accuracy.

**Query**
Input: query point $q \in \mathbb{R}^d$, number of bins to search $b$

1: Run inference on $M$ to compute $M(q)$.
2: Search for a near neighbor of $q$ in the bin $M(q)$, i.e., among the candidates $\{p \in P : M(p) = M(q)\}$.
3: If $M$ furthermore predicts a *distribution* over bins, search for a near neighbor in the $b$ top-ranked bins according to the ranking induced by the distribution (i.e., from the most likely bin to less likely ones).

---

Algorithm 1: Nearest neighbor search with a learned space partition

a distribution over the bin containing a point chosen uniformly at random among $S$ nearest neighbors of $p$ (including $p$ itself). Now, for a predicted distribution $\mathcal{Q} = (q_1, q_2, \ldots, q_m)$, we seek to minimize the KL divergence between $\mathcal{P}$ and $\mathcal{Q}$: $\sum_{i=1}^{m} p_i \log \frac{p_i}{q_i}$. Intuitively, soft labels help guide the neural network with information about multiple bin ranking. $S$ is a hyperparameter that needs to be tuned; we study its setting in the appendix (cf. Figure 6b).

# 3 Sparse hyperplane-induced cuts in $k$-NN graphs

We state and prove a theorem that shows, under certain mild assumptions, that the $k$-NN graph of a dataset $P \subseteq \mathbb{R}^d$ can be partitioned by a hyperplane such that the induced cut is sparse (i.e., has few crossing edges while the sizes of two parts are similar). The theorem is based on the framework of (Andoni et al., 2018b;c) and uses spectral techniques.

We start with some notation. Let $N_k(p)$ be the set of $k$ nearest neighbors of $p$ in $P$. The degree of $p$ in the $k$-NN graph is $\deg(p) = |N_k(p) \cup \{p' \in P \mid p \in N_k(p')\}|$. Let $\mathcal{D}$ be the distribution over the dataset $P$, where a point $p \in P$ is sampled with probability proportional to its degree $\deg(p)$. Let $\mathcal{D}_{\text{close}}$ be the distribution over pairs $(p, p') \in P \times P$, where $p \in P$ is uniformly random, and $p'$ is a uniformly random element of $N_k(p)$. Denote $\alpha = \mathrm{E}_{(p,p') \in \mathcal{D}_{\text{close}}}[\|p - p'\|_2^2]$ and $\beta = \mathrm{E}_{x_1 \sim \mathcal{D}, x_2 \sim \mathcal{D}}[\|p_1 - p_2\|_2^2]$. We will proceed assuming that $\alpha$ (typical distance between a data point and its nearest neighbors) is noticeably smaller than $\beta$ (typical distance between two independent data points).

The following theorem implies, informally speaking, that if $\alpha \ll \beta$, then there exists a hyperplane which splits the dataset into two parts of not too different size while separating only few pairs of $(p, p')$, where $p'$ is one of the $k$ nearest neighbors of $p$. For the proof of the theorem, see Appendix C.

**Theorem 3.1.** *There exists a hyperplane $H = \{x \in \mathbb{R}^d \mid \langle a, x \rangle = b\}$ such that the following holds. Let $P = P_1 \cup P_2$ be the partition of $P$ induced by $H$: $P_1 = \{p \in P \mid \langle a, p \rangle \leq b\}$, $P_2 = \{p \in P \mid \langle a, p \rangle > b\}$. Then, one has:*

$$\frac{\Pr_{(p,p') \sim \mathcal{D}_{\text{close}}}[p \text{ and } p' \text{ are separated by } H]}{\min\{\Pr_{p \sim \mathcal{D}}[p \in P_1], \Pr_{p \sim \mathcal{D}}[p \in P_2]\}} \leq \sqrt{\frac{2\alpha}{\beta}}. \tag{1}$$

# 4 Experiments

**Datasets** For the experimental evaluation, we use three standard ANN benchmarks (Aumüller et al., 2017): SIFT (image descriptors, 1M 128-dimensional points), GloVe (word embeddings (Pennington et al., 2014), approximately 1.2M 100-dimensional points, normalized), and MNIST (images of digits, 60K 784-dimensional points). All three datasets come with 10 000 query points, which are used for evaluation. We include the results for SIFT and GloVe in the main text, and MNIST in Appendix A.

**Evaluation metrics** We mainly investigate the trade-off between the number of candidates generated for a query point, and the $k$-NN accuracy, defined as the fraction of its $k$ nearest neighbors that are among those candidates. The number of candidates determines the processing time of an individual query. Over the entire query set, we report both the *average* as well as the $0.95$-*th quantile* of the number of candidates. The former measures the *throughput*[1] of the data structure, while the latter measures its *latency*.[2] We focus on parameter regimes that yield $k$-NN accuracy of at least 0.75, in the setting $k = 10$. Additional results with broader regimes of accuracy and of $k$ are included in the appendix.

**Our methods** We evaluate two variants of our method, with two different choices of the supervised learning component:

- **Neural LSH:** In this variant we use small neural networks. We compare this method with $k$-means clustering, Iterative Quantization (ITQ) (Gong et al., 2013), Cross-polytope LSH (Andoni et al., 2015), and Neural Catalyzer (Sablayrolles et al., 2019) composed over $k$-means clustering. We evaluate partitions into 16 bins and 256 bins. We test both one-level (non-hierarchical) and two-level (hierarchical) partitions. Queries are multi-probe.
- **Regression LSH:** This variant uses logistic regression as the supervised learning component and, as a result, produces very simple partitions induced by *hyperplanes*. We compare this method with PCA trees (Sproull, 1991; Kumar et al., 2008; Abdullah et al., 2014), random projection trees (Dasgupta & Sinha, 2013), and recursive bisections using 2-means clustering. We build trees of hierarchical bisections of depth up to 10 (thus total number of leaves up to 1024). The query procedure descends a single root-to-leaf path and returns the candidates in that leaf.

## 4.1 Implementation details

Neural LSH uses a fixed neural network architecture for the top-level partition, and a fixed architecture for all second-level partitions. Both architectures consist of several blocks, where each block is a fully-connected layer + batch normalization (Ioffe & Szegedy, 2015) + ReLU activations. The final block is followed by a fully-connected layer and a softmax layer. The resulting network predicts a distribution over the bins of the partition. The only difference between the top-level network the second-level network architecture is their number of blocks ($b$) and the size of their hidden layers ($s$). In the top-level network we use $b = 3$ and $s = 512$. In the second-level networks we use $b = 2$ and $s = 390$. To reduce overfitting, we use dropout with probability 0.1 during training. The networks are trained using the Adam optimizer (Kingma & Ba, 2015) for under 20 epochs on both levels. We reduce the learning rate multiplicatively at regular intervals. The weights are initialized with Glorot initialization (Glorot & Bengio, 2010). To tune soft labels, we try different values of $S$ between 1 and 120.

We evaluate two settings for the number of bins in each level, $m = 16$ and $m = 256$ (leading to a total number of bins of the total number of bins in the two-level experiments are $16^2 = 256$ and $256^2 = 65\,536$, respectively). In the two-level setting with $m = 256$ the bottom level of Neural LSH uses $k$-means instead of a neural network, to avoid overfitting when the number of points per bin is tiny. The other configurations (two-levels with $m = 16$ and one-level with either $m = 16$ or $m = 256$) we use Neural LSH at all levels.

We slightly modify the KaHIP partitioner to make it more efficient on the $k$-NN graphs. Namely, we introduce a hard threshold of 2000 on the number of iterations for the local search part of the algorithm, which speeds up the partitioning dramatically, while barely affecting the quality of the resulting partitions.

---

[1]Number of queries per second.

[2]Maximum time per query, modulo a small fraction of outliers.

|            |          | GloVe | | SIFT | |
|------------|----------|----------|----------------|----------|----------------|
|            |          | Averages | 0.95-quantiles | Averages | 0.95-quantiles |
| **One level** | 16 bins  | 1.745 | 2.125 | 1.031 | 1.240 |
|            | 256 bins | 1.491 | 1.752 | 1.047 | 1.348 |
| **Two levels** | 16 bins  | 2.176 | 2.308 | 1.113 | 1.306 |
|            | 256 bins | 1.241 | 1.154 | 1.182 | 1.192 |

Figure 3: Largest ratio between the number of candidates for Neural LSH and $k$-means over the settings where both attain the same target 10-NN accuracy, over accuracies of at least 0.85. See details in Section 4.2.

## 4.2 Comparison with multi-bin methods

Figure 4 shows the empirical comparison of Neural LSH with $k$-means clustering, ITQ, Cross-polytope LSH, and Neural Catalyzer composed over $k$-means clustering. It turns out that $k$-means is the strongest among these baselines.[3] The points depicted in Figure 4 are those that attain accuracy $\geq 0.75$. In the appendix (Figure 10) we include the full accuracy range for all methods.

In all settings considered, Neural LSH yields consistently better partitions than $k$-means.[4] Depending on the setting, $k$-means requires significantly more candidates to achieve the same accuracy:

- Up to $117\%$ more for the average number of candidates for GloVe;

- Up to $130\%$ more for the 0.95-quantiles of candidates for GloVe;

- Up to $18\%$ more for the average number of candidates for SIFT;

- Up to $34\%$ more for the 0.95-quantiles of candidates for SIFT;

Figure 3 lists the largest multiplicative advantage in the number of candidates of Neural LSH compared to $k$-means, for accuracy values of at least 0.85. Specifically, for every configuration of $k$-means, we compute the ratio between the number of candidates in that configuration and the number of candidates of Neural LSH in its optimal configuration, among those that attained at least the same accuracy as that $k$-means configuration.

We also note that in all settings except two-level partitioning with $m = 256$,[5] Neural LSH produces partitions for which the 0.95-quantiles for the number of candidates are very close to the average number of candidates, which indicates very little variance between query times over different query points. In contrast, the respective gap in the partitions produced by $k$-means is much larger, since unlike Neural LSH, it does not directly favor balanced partitions. This implies that Neural LSH might be particularly suitable for latency-critical NNS applications.

**Model sizes.** The largest model size learned by Neural LSH is equivalent to storing about $\approx 5700$ points for SIFT, or $\approx 7100$ points for GloVe. This is considerably larger than $k$-means with $k \leq 256$, which stores at most 256 points. Nonetheless, we believe the larger model size is acceptable for Neural LSH, for the following reasons. First, in most of the NNS applications, especially for the distributed setting, the bottleneck in the high accuracy regime is the memory accesses needed to retrieve candidates and the further processing (such as distance computations, exact or approximate). The model size is not a hindrance as long as does not exceed certain reasonable limits (e.g., it should fit into a CPU cache). Neural LSH significantly reduces the memory access cost, while increasing the model size by an acceptable amount. Second, we have observed that the quality of the Neural LSH partitions is not too sensitive to decreasing the sizes the hidden layers. The model sizes we report are, for the sake of concreteness, the largest ones that still lead to improved performance. Larger models do not increase the accuracy, and sometimes decrease it due to overfitting.

---

[3]It is important to note that ITQ is not designed to produce space partitions; as explained in Section 1, it does so as a side-effect. Simiarly, Neural Catalyzer is not designed to enhance partitions. The comparison is intended to show that they do not outperform indexing techniques despite being outside their intended application.

[4]We note that two-level partitioning with $m = 256$ is the best performing configuration of $k$-means, for both SIFT and GloVe, in terms of the minimum number of candidates that attains 0.9 accuracy. Thus we evaluate this baseline at its optimal performance.

[5]As mentioned earlier, in this setting Neural LSH uses $k$-means at the second level, due to the large overall number of bins compared to the size of the datasets. This explains why the gap between the average and the 0.95-quantile number of candidates of Neural LSH is larger for this setting.

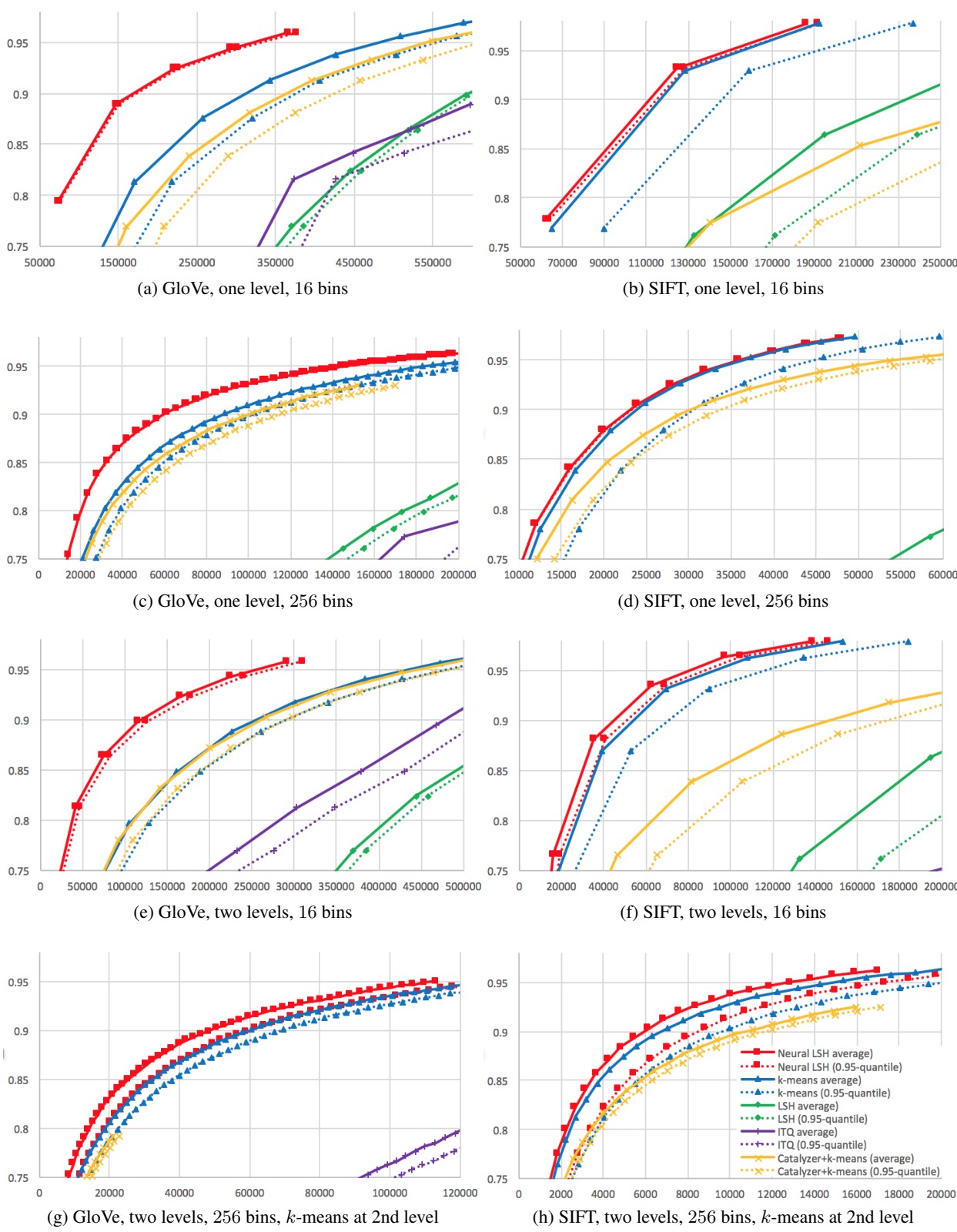

Figure 4: Comparison of Neural LSH with baselines; x-axis is the number of candidates, y-axis is the 10-NN accuracy

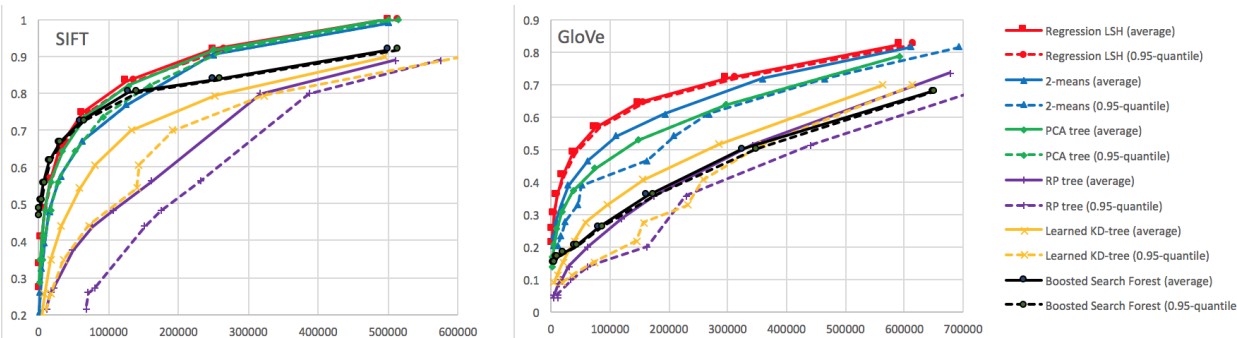

Figure 5: Comparison of decision trees built from hyperplanes: x-axis – number of candidates, y-axis – 10-NN accuracy

## 4.3 Comparison with tree-based methods

Next we compare binary decision trees, where in each tree node a *hyperplane* is used to determine which of the two subtrees to descend into. We generate hyperplanes with the following methods: Regression LSH, the Learned KD-tree of Cayton & Dasgupta (2007), the Boosted Search Forest of Li et al. (2011), cutting the dataset into two equal halves along the top PCA direction (Sproull, 1991; Kumar et al., 2008), 2-means clustering, and random projections of the centered dataset (Dasgupta & Sinha, 2013; Keivani & Sinha, 2018). We build trees of depth up to $10$, which correspond to hierarchical partitions with the up to $2^{10} = 1024$ bins. Results for GloVe and SIFT are summarized in Figure 5 (see appendix). For random projections, we run each configuration 30 times and average the results.

For GloVe, Regression LSH significantly outperforms 2-means, while for SIFT, Regression LSH essentially matches 2-means in terms of the *average* number of candidates, but shows a noticeable advantage in terms of the 0.95-percentiles. In both instances, Regression LSH significantly outperforms PCA tree, and all of the above methods dramatically improve upon random projections.

Note, however, that random projections have an additional benefit: in order to boost search accuracy, one can simply repeat the sampling process several times and generate an ensemble of decision trees instead of a single tree. This allows making each individual tree relatively deep, which decreases the overall number of candidates, trading space for query time. Other considered approaches (Regression LSH, 2-means, PCA tree) are inherently deterministic, and boosting their accuracy requires more care: for instance, one can use partitioning into blocks as in Jégou et al. (2011), or alternative approaches like Keivani & Sinha (2018). Since we focus on individual partitions and not ensembles, we leave this issue out of the scope.

## 4.4 Additional experiments

In this section we include several additional experiments.

First, we study the effect of setting $k$. We evaluate the 50-NN accuracy of Neural LSH when the partitioning step is run on either the 10-NN or the 50-NN graph.[6] We compare both algorithms to $k$-means with $k = 50$. Figure 6a compares these three algorithms on GloVe for 16 bins reporting average numbers of candidates. From this plot, we can see that for $k = 50$, Neural LSH convincingly outperforms $k$-means, and whether we use 10-NN or 50-NN graph matters very little.

Second, we study the effect of varying $S$ (the soft labels parameter) for Neural LSH on GloVe for 256 bins. See Figure 6b where we report the average number of candidates. As we can see from the plot, the setting $S = 15$ yields much better results compared to the vanilla case of $S = 1$. However, increasing $S$ beyond 15 brings diminishing returns on the overall accuracy.

---

[6]Neural LSH can solve $k$-NNS by partitioning the $k'$-NN graph, for any $k, k'$; they do not have to be equal.

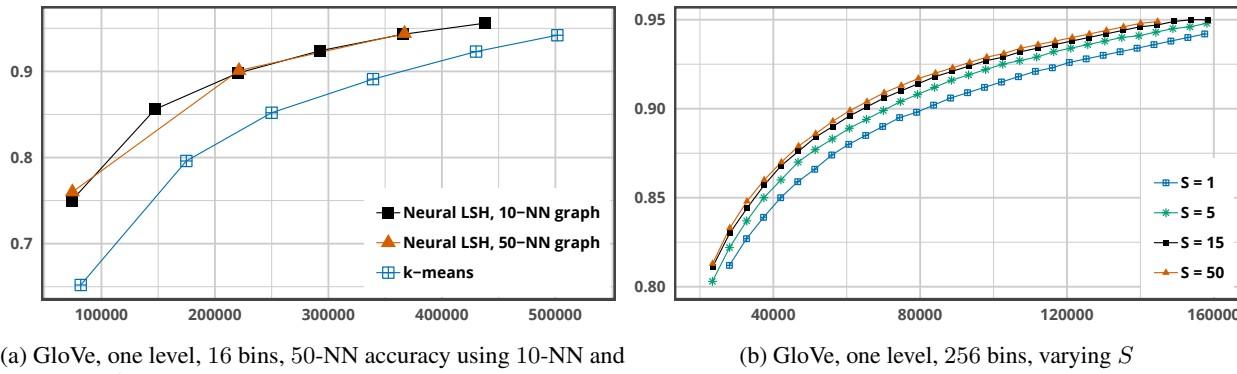

(a) GloVe, one level, 16 bins, 50-NN accuracy using 10-NN and 50-NN graphs

(b) GloVe, one level, 256 bins, varying $S$

Figure 6: Effect of various hyperparameters

# 5 Conclusions and future directions

We presented a new technique for finding partitions of $\mathbb{R}^d$ which support high-performance indexing for sublinear-time NNS. It proceeds in two major steps: (1) We perform a combinatorial balanced partitioning of the $k$-NN graph of the dataset; (2) We extend the resulting partition to the whole ambient space $\mathbb{R}^d$ by using supervised classification (such as logistic regression, neural networks, etc.). Our experiments show that the new approach consistently outperforms quantization-based and tree-based partitions. There is a number of exciting open problems we would like to highlight:

- Can we use our approach for NNS over *non-Euclidean* geometries, such as the edit distance (Zhang & Zhang, 2017) or the optimal transport distance (Kusner et al., 2015)? The graph partitioning step directly carries through, but the learning step may need to be adjusted.
- Can we jointly optimize a graph partition *and* a classifier at the same time? By making the two components aware of each other, we expect the quality of the resulting partition of $\mathbb{R}^d$ to improve. A related approach has been successfully applied in Li et al. (2011) for hyperplane tree partitions.
- Can our approach be extended to learning *several* high-quality partitions that complement each other? Such an ensemble might be useful to trade query time for memory usage (Andoni et al., 2017).
- Can we use machine learning techniques to improve *graph-based* indexing techniques (Malkov & Yashunin, 2018) for NNS? (This is in contrast to partition-based indexing, as done in this work).
- Our framework is an example of combinatorial tools aiding "continuous" learning techniques. A more open-ended question is whether other problems can benefit from such symbiosis.

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

# A    Results for MNIST

We include experimental results for the MNIST dataset, where all the experiments are performed exactly in the same way as for SIFT and GloVe. Consistent with the trend we observed for SIFT and GloVe, Neural LSH consistently outperforms $k$-means (see Figure 7) both in terms of average number of candidates and especially in terms of the $0.95$-th quantiles. We also compare Regression LSH with recursive 2-means, as well as PCA tree and random projections (see Figure 8), where Regression LSH consistently outperforms the other methods.

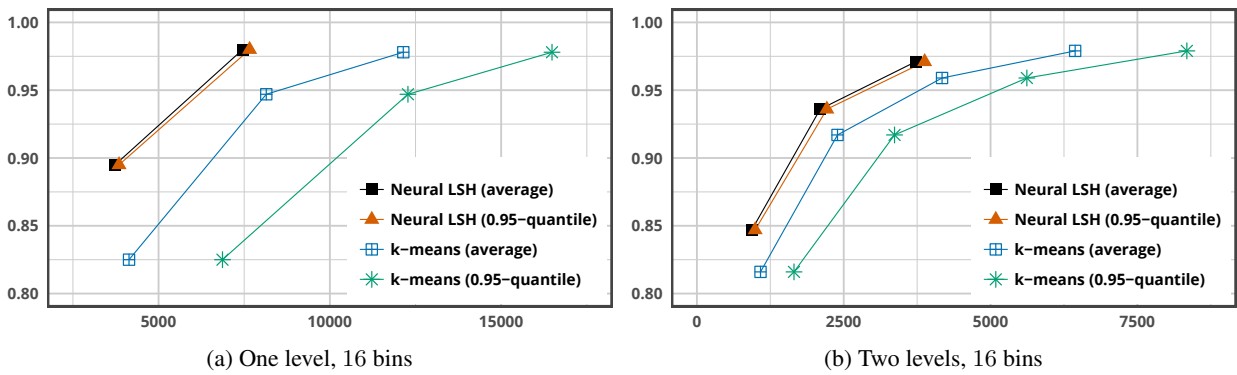

(a) One level, 16 bins                    (b) Two levels, 16 bins

Figure 7: MNIST, comparison of Neural LSH with $k$-means; x-axis – number of candidates, y-axis – 10-NN accuracy

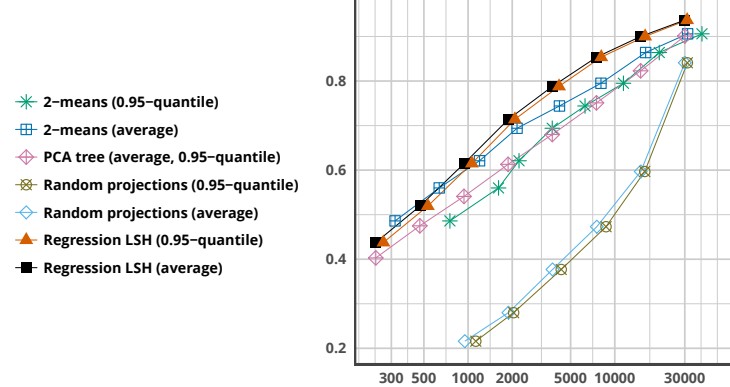

Figure 8: MNIST, comparison of trees built from hyperplanes; x-axis – number of candidates, y-axis – 10-NN accuracy

# B    Effect of Neural Catalyzer on Space Partitions

In this section we compare vanilla $k$-means with $k$-means run after applying a Neural Catalyzer map (Sablayrolles et al., 2019). The goal is to check whether the Neural Catalyzer – which is designed to boost up the performance of sketching methods for NNS by adjusting the input geometry – could also improve the quality of space partitions for NNS. See Figure 9 for the comparison on GloVe and SIFT with 16 bins. On both datasets (especially SIFT), Neural Catalyzer in fact degrades the quality of the partitions. We observed a similar trend for other numbers of bins than the setting reported here. These findings support our observation that while both indexing and sketching for NNS can benefit from learning-based enhancements, they are fundamentally different approaches and require different specialized techniques.

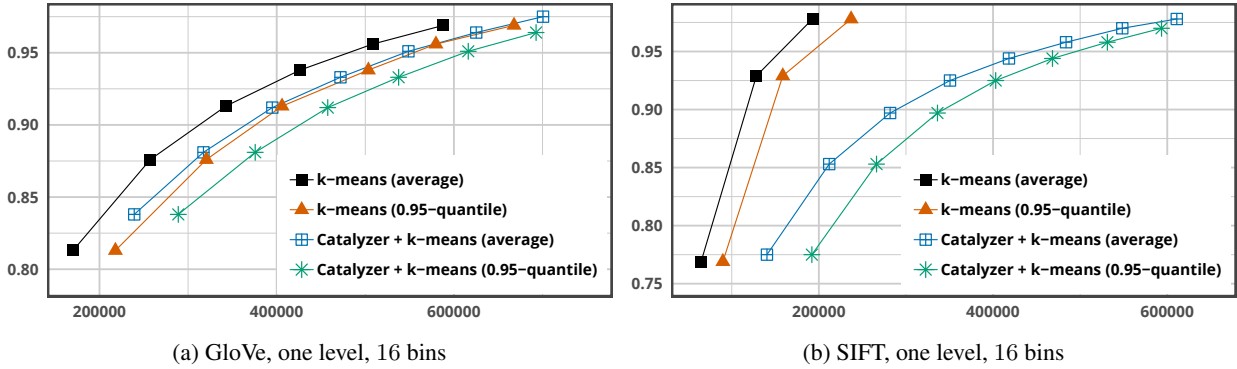

(a) GloVe, one level, 16 bins                      (b) SIFT, one level, 16 bins

Figure 9: Comparison of $k$-means and Catalyzer + $k$-means

# C    Proof of Theorem 3.1

*Proof.* Consider an undirected graph $G = (V, E)$, where the set of vertices $V$ is $P$, and the (multi-)set of edges contains an edge $(p, p')$ for every $p' \in N_k(p)$. The graph contains $n$ vertices and $kn$ edges, and some of the edges might be double (if $p' \in N_k(p)$ and $p \in N_k(p')$ at the same time). Let $A_G$ be the symmetric adjacency matrix of $G$ normalized by $2kn$ (so that the sum of all the entries equals to 1, thus giving a probability distribution over $P \times P$, which can be seen to be equal to $\mathcal{D}_{\text{close}}$). The rows and columns of $A_G$ can naturally be indexed by the points of $P$. Denote $\rho_G(p) = \sum_{p'} (A_G)_{p,p'}$. It is immediate to check that $\rho_G$ yields a distribution over $P$, which can be seen to be equal to $\mathcal{D}$. Denote $D_G = \text{diag}(\rho_G)$. Denote $L_G = D_G - A_G$ the Laplacian of $A_G$. Due to the equivalence of $\rho_G$ and $\mathcal{D}$ and $A_G$ and $\mathcal{D}_{\text{close}}$, we have:

$$\frac{\alpha}{\beta} = \frac{\sum_{p,p' \in P} (A_G)_{p,p'} \cdot \|p - p'\|_2^2}{\sum_{p,p' \in P} \rho_G(p)\rho_G(p') \cdot \|p - p'\|_2^2}. \tag{2}$$

By considering all possible coordinate projections and using additivity of $\| \cdot \|_2^2$ over coordinates, we conclude that there exists a coordinate $i^* \in [d]$ such that:

$$\frac{\sum_{p,p' \in P} (A_G)_{p,p'} \cdot (p_{i^*} - p'_{i^*})^2}{\sum_{p,p' \in P} \rho_G(p)\rho_G(p') \cdot (p_{i^*} - p'_{i^*})^2} \leq \frac{\alpha}{\beta}. \tag{3}$$

Define a vector $y \in \mathbb{R}^P$ by $y_p = p_{i^*}$. We now apply the following standard fact from spectral graph theory: If $A$ is the weighted adjacency matrix of a graph, and $L$ is its Laplacian matrix, then $x^t L x = \sum_{i,j=1}^{n} A_{ij}(x_i - x_j)^2$ for all $x \in \mathbb{R}^n$. Thus the numerator of (3) becomes $y^t L_G y$. For the denominator, consider the graph $H$ on $P$ in which every pair $p, p'$ is connected by an edge of weight $\rho_G(p)\rho_G(p')$.

- Its weighted adjacency matrix $A_H$ is given by $(A_H)_{p,p'} = \rho_G(p)\rho_G(p')$ for $p \neq p'$, and with zeros on the diagonal. Thus $A_H = \rho_G \rho_G^t - D_G^2$ (recall that $D_G = \text{diag}(\rho_G)$).

- The degree of each node $p$ in $H$ equals $\rho_G(p) \sum_{p' \in P \setminus \{p\}} \rho_G(p') = \rho_G(p) - (\rho_G(p))^2$ (recall that $\sum_{p' \in P} \rho_G(p') = 1$). Therefore the diagonal degree matrix of $H$ is $D_H = D_G - D_G^2$.

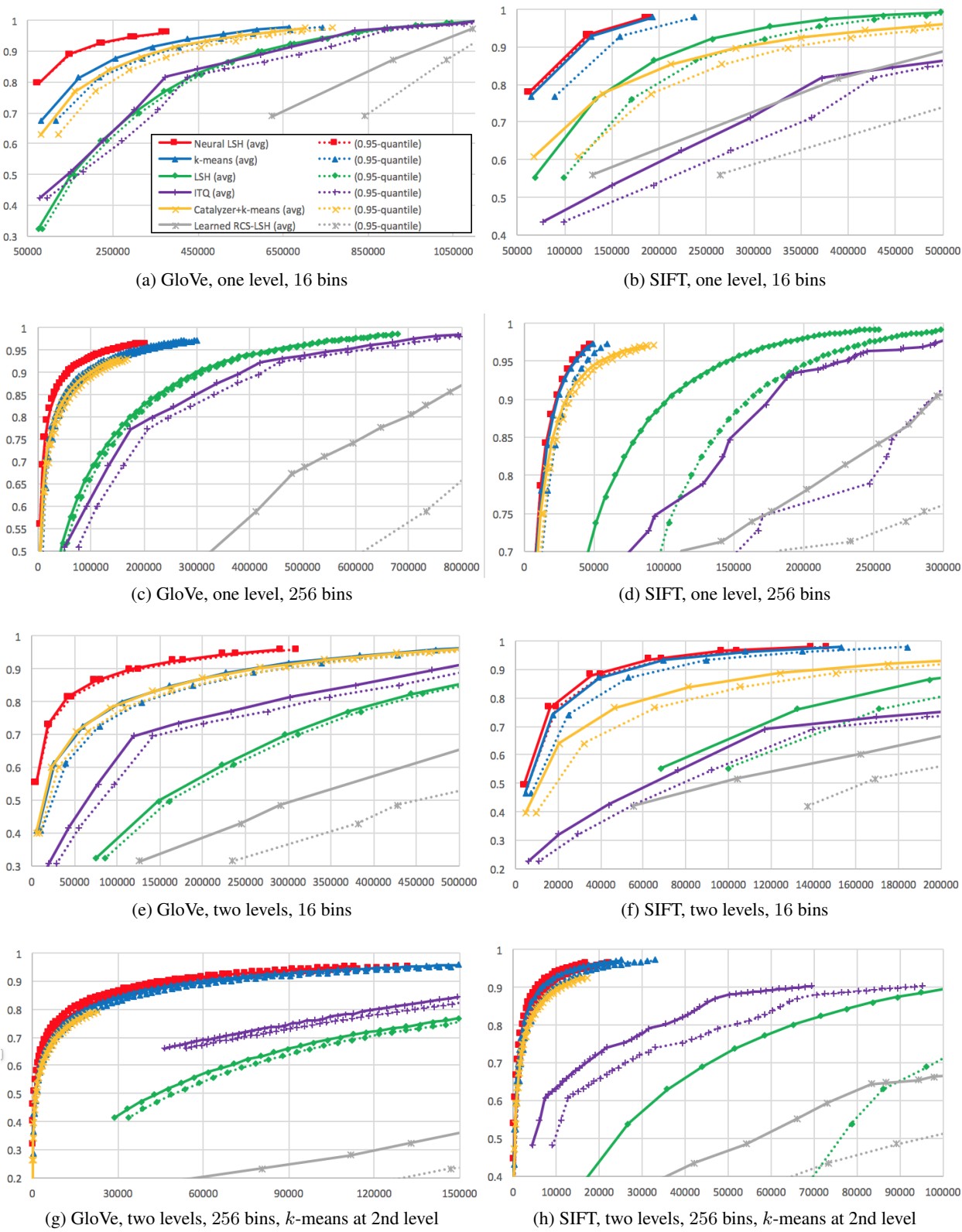

(a) GloVe, one level, 16 bins

(b) SIFT, one level, 16 bins

(c) GloVe, one level, 256 bins

(d) SIFT, one level, 256 bins

(e) GloVe, two levels, 16 bins

(f) SIFT, two levels, 16 bins

(g) GloVe, two levels, 256 bins, $k$-means at 2nd level

(h) SIFT, two levels, 256 bins, $k$-means at 2nd level

Figure 10: Results from Figure 4 with broader candidate and accuracy regimes. The "Learned RCS-LSH" baseline is the learned rectilinear cell structure locality sensitive hashing method of Cayton & Dasgupta (2007).

Together, the Lapacian of $H$ is $L_H = D_H - A_H = D_G - \rho_G \rho_G^t$. Therefore the denominator of (3) becomes $y^t(D_G - \rho_G \rho_G^t)y$. Overall, we have:

$$\frac{y^t L_G y}{y^t(D_G - \rho_G \rho_G^t)y} \leq \frac{\alpha}{\beta}.$$

Next, we define $\widetilde{y} = y - c \cdot \mathbf{1}$, where $\mathbf{1}$ is the all-1's vector, and $c$ is the scalar $c = (y^t \rho_G)/(\mathbf{1}^t \rho_G)$. This scalar is chosen to render $\widetilde{y} \perp \rho_G$. Furthermore, since $\mathbf{1}$ is in the kernel of every Laplacian matrix, we have $L_G \widetilde{y} = L_G y$ and $L_H \widetilde{y} = L_H y$. Together, we get

$$\frac{\widetilde{y}^t L_G \widetilde{y}}{\widetilde{y}^t D_G \widetilde{y}} = \frac{y^t L_G y}{y^t(D_G - \rho_G \rho_G^t)y} \leq \frac{\alpha}{\beta},$$

Now by the Cheeger's inequality (Chung, 1996), we conclude that there exists a threshold $y_0 \in \mathbb{R}$ such that:

$$\frac{\sum_{p_1,p_2:\widetilde{y}_{p_1} \leq y_0, \widetilde{y}_{p_2} > y_0}(A_G)_{p_1,p_2}}{\min\{\sum_{p:\widetilde{y}_p \leq y_0} \rho_G(p), \sum_{p:\widetilde{y}_p > y_0} \rho_G(p)\}} \leq \sqrt{2 \cdot \frac{\widetilde{y}^t L_G \widetilde{y}}{\widetilde{y}^t D_G \widetilde{y}}} \leq \sqrt{\frac{2\alpha}{\beta}}. \tag{4}$$

One can trace back all the definitions and observe that the set $\{p \in P \colon \widetilde{y}_p \leq y_0\}$ is induced by an (axis-aligned) hyperplane, and the left-hand side of (4) is nothing else but the left-hand side of (1). $\qquad \square$

