# OpenReview forum: "Learning Space Partitions for Nearest Neighbor Search"
_ICLR.cc/2020/Conference — Accept (Poster)_

### Official Review · AnonReviewer3 · 2019-10-17
**Official Blind Review #3**

**Rating:** 3

**Review:**

This paper proposes a technique for partitioning (presumably large number of) points in a d-dimensional, such that approximate nearest neighbor search in this space can be efficiently performed by reducing to search in a smaller partition. To this end, the paper provides the following procedure:
1. Build an exact k nearest neighbor graph of the provided dataset.
2. Partition this graph into m balanced sets while minimize inter-partition edges.
3. Train a neural net or linear model to predict the partition for each point. They also propose creating hierarchical partitions in the same way and also propose training with "soft" partition labels.

At test time, the trained model is used to predict probably partitions which are then searched over for nearest neighbors. The authors compare this technique to, most notably, a k-means based technique where centroids are used to partition the space. They show that on Glove and SIFT datasets, the neural partitioning approach is more accurate i.e. it can correctly identify correct nearest neighbors significantly more than k-means.

######
Pros:
1. This proposed approach of using neural techniques to build partitions of space for fast nearest neighbors is novel. While k-means-style techniques have been proposed to learn index structures, this is the first application of neural techniques as far as I can tell.
2. The paper is well-written and is easy to read. The experiments are sufficiently detailed and well documented.

Cons:
1. Experiments: The paper reports the number of correct neighbors obtained while searching on average and at 0.95 quantile. However, the core motivation of the paper is *fast* nearest neighbors search. In that vein, the metric of interest must incorporate time e.g. queries per second at a fixed level of accuracy. This is not new -- e.g. the benchmarks provided by Andoni and co-authors are evaluated on a QPS basis (https://github.com/erikbern/ann-benchmarks). The same holds for Amueller et al.
2. Baselines: A related point is, what happens to baseline algorithms when they are given more "capacity". E.g. comparing with k-means baselines with a significantly large value of k or a multi-level k-means algorithm (e.g. considered by "FAST APPROXIMATE NEAREST NEIGHBORS WITH AUTOMATIC ALGORITHM CONFIGURATION" by Muje and Lowe). The comparison in the paper with k-means baseline with k=50 does not seem fair here. E.g. a hierarchical neural LSH approach with 256 partitions at top and 10 k means partitions at the second level performs 2560 distance computations. Alternatively, the authors can directly compare QPS as stated above.
3. The proof provided in the appendix is not entirely clear. E.g. after step 3, how does the next step (before (4)) follow? It is easy to see the denominator but the numerator is not clear.


Suggestions:
1. Please cite the following paper which is highly related and highly cited: FAST APPROXIMATE NEAREST NEIGHBORS WITH AUTOMATIC ALGORITHM CONFIGURATION.


**Experience Assessment:**

I do not know much about this area.

**Review Assessment: Checking Correctness Of Derivations And Theory:**

I carefully checked the derivations and theory.

**Review Assessment: Checking Correctness Of Experiments:**

I carefully checked the experiments.

**Review Assessment: Thoroughness In Paper Reading:**

I read the paper at least twice and used my best judgement in assessing the paper.

---

> ### Author Response · Authors · 2019-11-15
> **Response to Review #3**
>
> We thank the reviewer for the feedback and comments.
>
> > the metric of interest must incorporate time e.g. queries per second
>
> As we discuss in the introduction, the space partitioning approach to nearest neighbor search is widely applicable in several settings where other approaches are less suitable, like distributed architectures, secure computation and computation on a GPU. Since the "wall clock" running time depends on the specific setting/architecture, we decided to focus on architecture-independent metrics.
>
> > what happens to baseline algorithms when they are given more "capacity"
>
> Indeed, this is an important point. We address it at the bottom of page 6. Specifically, we believe the larger model size is acceptable for Neural LSH, since in many target applications (e.g., the distributed setting) the bottleneck in the high accuracy regime is the memory accesses needed to retrieve candidates and the further processing. The model size is not a hindrance as long as does not exceed certain reasonable limits, e.g., it should fit into a CPU cache.
>
> We also note that in the aforementioned scenarios the number of partitions is defined by the architecture of the system and cannot be adjusted/optimized. E.g., in the distributed setting, the number of partitions is equal to the number of machines that the data is distributed to.
>
> > Please cite the following paper
>
> Indeed, this is a highly relevant paper, thank you for pointing this out. We included citations to it wherever we mention prior work on nearest neighbor using k-means.
>
> > The proof provided in the appendix is not entirely clear.
>
> Indeed, those steps in the proof have been overly compressed, and we apologize for the lack of clarity. We have expanded the proof in the paper.

---

### Official Review · AnonReviewer1 · 2019-10-22
**Official Blind Review #1**

**Rating:** 6

**Review:**


The paper proposes a scheme to learn space partitions for improved
nearest neighbor search by first converting the search problem into a
supervised classification problem by graph-partitioning the
k-nearest-neighbor graph, and then using some machine learning model
to learn space partitions corresponding to the supervised problem. The
scheme is theoretically motivated and the empirical results
demonstrates the utility of the proposed scheme against various
baselines.


While the paper presents a promising new direction for
nearest-neighbor search with space partitioning schemes, I am somewhat
on the border, leaning towards reject. This is mostly due to my lack
of understanding why Cayton & Dasgupta (2007) and Li et al. (2011) are
not valid baselines and are dismissed as "hyperplane
partitions". It would be very helpful to me as a reader to understand
why this literature does not warrant to be a valid
baseline. Especially given the following connection points:

- Firstly, the theoretical results presented by the authors also
  limit to hyperplane splits.
- Secondly, the authors present empirical evaluation of recursive
  hyperplane partitions where the proposed scheme is the only learning
  based scheme while the techniques presented in Cayton & Dasgupta
  (2007) and Li et al. (2011) would have been the learning based
  partitioning baseline to beat.
- Thirdly, LSH space partitioning is considered as a baseline in
  Section 4.2 with the Neural LSH and Neural Catalyzer + k-means being
  the only learning based baseline; Cayton & Dasgupta (2007)'s learned
  LSH scheme would have been a valid baseline to the best of my
  understanding.
- Finally, the authors mention the need for coupling the graph
  partitioning and the supervised learning phase. Li et al., 2011
  utilizes a loss function that essentially promotes a sparse graph
  cut while preserving near-neighborhoods, providing the closely
  coupled scheme. In the original paper, the authors utilize a
  recursive hyperplane partitioning scheme. However, a neural network
  with a softmax layer can be trained to minimize this closely coupled
  loss directly, obviating the need for the separate graph
  partitioning and supervised learning phases proposed in this
  paper. But it is possible that I have misinterpreted the connection
  and any clarification here would be very helpful.

Minor:

- It would be very helpful as a reader to understand how sensitive the
  neural network based partioning scheme is to the network
  architecture and hyperparameters. Some intuition behind the choices
  would be very useful.
- It is a little unclear how the plots in Figure 2 are generated --
  for each query, we would have access to the accuracy vs. number of
  candidates curve. In that case, it is unclear how the curves are
  aggregated across queries to generate average number of candidates
  (and 0.95 quantile) vs. accuracy curves. Given that not all queries
  in all the methods (including the proposed ones) will have the same
  sequences of accuracies, the quantile curve is especially unclear.
- The authors mention that neural LSH is only useful when the bin
  sizes are large, and otherwise k-means is more useful. Is there a
  straightforward way to decide when to switch between the two? Is it
  another hyperparameter we need to tune over while generating these
  space partitions?
- Theorem 3.1 considers only hyperplane partitions while neural
  networks usually generate nonlinear partitions. What are the
  conditions on the nonlinear splits that allows the guarantees to
  improve (or at least not degrade)? Is it obviously always improved?


**Experience Assessment:**

I have published in this field for several years.

**Review Assessment: Checking Correctness Of Derivations And Theory:**

I assessed the sensibility of the derivations and theory.

**Review Assessment: Checking Correctness Of Experiments:**

I carefully checked the experiments.

**Review Assessment: Thoroughness In Paper Reading:**

I read the paper thoroughly.

---

> ### Author Response · Authors · 2019-11-15
> **Response to Review #1**
>
> We thank you for your feedback and comments. Indeed, both Cayton & Dasgupta (2007) and Li et al. (2011) are valid baselines. We have contacted the authors of both works, but unfortunately, there are no publicly available implementation. We implemented and tuned these algorithms in the rebuttal timeframe, and produced results in the following settings:
>
> For binary tree partitions, we include the results of Li et al. (2011) and Cayton & Dasgupta (2007) KD-tree on the SIFT dataset. We will include GloVe and MNIST in the final version. Link: https://anonymous.4open.science/repository/cdd789a8-818c-4675-98fd-39f8da656129/new_plots/twopart_sift.png
>
> For multi-way partitions, we include the results of Li et al. (2011) and Cayton & Dasgupta (2007) RCS-LSH on both SIFT and GloVe, with 1 level (both 16 and 256 parts). We will include the 2-level experiments, and the results for MNIST, in the final version. Links:
> https://anonymous.4open.science/repository/cdd789a8-818c-4675-98fd-39f8da656129/new_plots/multipart_sift_1_16.png
> https://anonymous.4open.science/repository/cdd789a8-818c-4675-98fd-39f8da656129/new_plots/multipart_glove_1_256.png
> https://anonymous.4open.science/repository/cdd789a8-818c-4675-98fd-39f8da656129/new_plots/multipart_sift_1_16.png
> https://anonymous.4open.science/repository/cdd789a8-818c-4675-98fd-39f8da656129/new_plots/multipart_sift_1_256.png
>
> We would like to highlight the following points:
> Our approach, while being learning-based, differs drastically from both of these works.
> Regarding the learned RCS/LSH algorithm of Cayton & Dasgupta (2007): It indeed fits into the comparison to Neural LSH as it produces a multi-part partition. However, that paper mentions that they do not observe improvement over classic LSH (which is included in our experiments) on MNIST, due to the uniformity of random projections - a property shared also by SIFT and GloVe. Nonetheless we will include the baseline to draw a complete picture.
> Re:“the authors mention the need for coupling the graph partitioning and the supervised learning phase”: Our algorithm makes no such coupling; we mentioned this as a possible avenue of improvement in future work. We thank you for bringing to our attention that the same approach has been utilized by Li et al. (2011), and have added this to the discussion in Section 5. Since Li et al. (2011) do not discuss or experiment with neural networks, it is hard to say a priori how would combining this idea into their framework would perform, and this is an exciting area for future work.
>
> We are hoping that the above experiments address the main concern from your review. We will continue running the algorithms to evaluate their performance in all of our experimental settings, and will include the results in the final version of the paper.

---

> > ### Author Response · Authors · 2019-11-15
> > **Continued response**
> >
> > In what follows we address the comments in the “Minor” section of your review.
> >
> > > It would be very helpful as a reader to understand how sensitive the neural network based partitioning scheme is to the network architecture and hyperparameters. Some intuition behind the choices would be very useful.
> >
> > Through experimenting with various network architectures, we found that the partitioning results are not sensitive to the network architecture and many of the hyperparameters. Therefore, in choosing our network architecture, we optimized for the size of the neural networks to be small, for fast evaluation.
> > The effect of hyperparameters is studied in Figure 5 in appendix B. To summarize, the hyperparameter that the model is most sensitive to is S, the number of nearest neighbors whose bins are used as labels for training. Our experiments show that increasing S improves the accuracy noticeably, up to a certain point, and then the accuracy saturates (but does not degrade), cf. Fig 5(b).
> >
> >
> > > it is unclear how the curves are aggregated across queries to generate average number of candidates (and 0.95 quantile) vs. accuracy curves.
> >
> > To produce the plots we vary $m$, the number of bins searched (in multi-probe fashion) per query. That is, for each fixed m, we search for the nearest neighbor of each query in the m nearest bin to that query (including the one containing it). This gives us a number of candidates (total number of datapoints in those m bins) and a 0/1 success rate (whether the nearest neighbor is in those bins or not) per query. We average both of these (for a fixed $m$) over all queries to produce a single point on the curve. The curve is produced by gradually increasing $m$.
> >
> > > What are the conditions on the nonlinear splits that allows the guarantees to improve (or at least not degrade)? Is it obviously always improved?
> >
> > A priori we cannot give such a guarantee. In practice, we rely on the fact that the Cheeger inequality is often too pessimistic and much sparser cuts exist (not necessarily induced by hyperplanes). At the same time, if we assume that neural networks are “universal approximators”, then we expect them to capture such cut using not too many parameters.
> >
> >
> > > The authors mention that neural LSH is only useful when the bin sizes are large, and otherwise k-means is more useful. Is there a straightforward way to decide when to switch between the two? Is it another hyperparameter we need to tune over while generating these space partitions?
> >
> > Neural LSH is useful whenever there are sufficient data points per bin so the network does not overfit to the training set. A good way to determine when to use k-means instead is when the model achieves low accuracy on the validation set. Practically, through our experiments, we found that reducing the network capacity, adding regularizations to the loss, and dropouts are effective ways of avoiding overfitting, and making neural LSH outperform k-means even in cases where there are fewer than 100 points per bin.

---

### Official Review · AnonReviewer4 · 2019-10-22
**Official Blind Review #4**

**Rating:** 6

**Review:**

The authors propose a learnable space partitioning nearest-neighbour algorithm where the model learns a hierarchical space partition of the data (using graph partitioning) and learns to predict which buckets a query will reside in (using a learnable model).

The paper is an interesting combination of classical optimization and deep learning.

The authors make a concerted effort to compare to a range of space-partitioning-based nearest neighbour algorithms, however crucially none of these algorithms are state-of-the-art in comparison to some of the best PQ NNS or hierarchical Navigable Small World Graphs.

In terms of writing, the model description requires significant overhaul. I had to re-read it several times and sketch out what exactly you are doing. I would recommend an algorithm box to summarize both the hierarchical graph partitioning *and* the learnable query network.

4.3 "additional experiments" is some of the most interesting parts of the paper, I would recommend promoting these experimental results to the main paper.

I would recommend accepting this paper because I think it may lead to state-of-the-art NNS algorithms with theoretical guarantees, especially with a more powerful query network.

**Experience Assessment:**

I have published one or two papers in this area.

**Review Assessment: Checking Correctness Of Derivations And Theory:**

I did not assess the derivations or theory.

**Review Assessment: Checking Correctness Of Experiments:**

I assessed the sensibility of the experiments.

**Review Assessment: Thoroughness In Paper Reading:**

I read the paper at least twice and used my best judgement in assessing the paper.

---

> ### Author Response · Authors · 2019-11-15
> **Response to Review #4**
>
> Thank you for your feedback and comments.
>
> >> “none of these algorithms are state-of-the-art in comparison to some of the best PQ NNS or hierarchical Navigable Small World Graphs.”
> The space partitioning approach to nearest neighbor search is widely applicable in several settings where other approaches (like graph-based methods) are less suitable, like distributed architectures, secure computation and computation on a GPU. (This is elaborated on in the introduction.) Furthermore, sketching approaches (like PQ) are often combined with space partitions. These scenarios justify the study of space partitions on their own right, as demonstrated by the vast amount of existing (and ongoing) research on the subject.
>
> >> “the model description requires significant overhaul”
> We apologize for the lack of clarity. We will rewrite and section and add a stand-alone algorithm statement.
>
> >> “"additional experiments" is some of the most interesting parts of the paper, I would recommend promoting these experimental results to the main paper.”
>
> Indeed, this has a been a suboptimal concession to the page limit; we will add this content to the main text.
>
> For both of the latter comments, we defer the actual revision of the paper to the final version, as they would necessitate moving other material into the appendix due to the page limit.

---

### Decision · Program_Chairs · 2019-12-19

**Decision:**

Accept (Poster)

**Comment:**

This paper proposes a new framework for improved nearest neighbor search by learning a space partition of the data, allowing for better scalability in distributed settings and overall better performance over existing benchmarks.

The two reviewers who were most confident were both positive about the contributions and the revisions. The one reviewer who recommended reject was concerned about the metric used and whether comparison with baselines was fair. In my opinion, the authors seem to have been very receptive to reviewer comments and answered these issues to my satisfaction. After author and reviewer engagement, both R1 and myself are satisfied with the addition of the new baselines and think the authors have sufficiently addressed the major concerns. For the final version of the paper, I’d urge the authors to take seriously R4’s comment regarding clarity and add algorithmic details as per their suggestion.